# Influence of Load Knowledge on Biomechanics of Asymmetric Lifting

**DOI:** 10.3390/ijerph19063207

**Published:** 2022-03-09

**Authors:** Junshi Liu, Xingda Qu, Yipeng Liu

**Affiliations:** 1Institute of Human Factors and Ergonomics, Shenzhen University, Shenzhen 518060, China; jsliu@szu.edu.cn (J.L.); a184401661@foxmail.com (Y.L.); 2Key Laboratory of Optoelectronic Devices and Systems of Ministry of Education and Guangdong Province, Shenzhen University, Shenzhen 518060, China

**Keywords:** biomechanics, asymmetric lifting, load knowledge, low back pain

## Abstract

Background: Load knowledge has been identified as a factor affecting the risk of low back pain (LBP) during symmetric lifting. However, the effects of load knowledge in asymmetric lifting tasks have not been reported yet. The purpose of this study was to investigate the load knowledge influence on lifting biomechanics in asymmetric lifting tasks; Methods: Twenty-four male adults were recruited to complete a psychophysical lifting capacity test and a simulated asymmetric lifting task. The lifting task was set with load knowledge of ‘no knowledge’ (NK), ‘weight known’ (WK), ‘fragile material known’ (FK), and ‘weight and fragile material known’ (WFK) for different lifting load weights. Trunk kinematics and kinetics were collected and analyzed; Results: When fragility information was presented, trunk sagittal flexion acceleration, lateral flexion velocity and acceleration, and average lateral bending moment were significantly lowered at the deposit phase. Lifting a high load weight was found to significantly increase low back sagittal bending moment at the lifting phase and low back moments of all three dimensions at the deposit phase; Conclusions: The decrease of trunk kinematic load suggests that providing material fragility information to workers in asymmetric lifting tasks would be effective in reducing their risk of LBP.

## 1. Introduction

Low back pain (LBP) is a worldwide known musculoskeletal disorder. It is commonly seen that people who suffered from LBP routinely performed physically demanding work involving manual lifting tasks [1]. In occupational settings, active workers from the food-line industry, warehouse, airport, construction, and hospital were mainly reported to have LBP with sick leave [2,3,4]. There comes a tremendous medical care cost associated with LBP, which was estimated over millions of dollars each year based on a report received from a sample of the USA [5]. The cause of this problem was considered to start from the detrimental biomechanical effect and following with psychophysical factors to further deteriorate health [6]. Biomechanical factors related to LBP in the manual lifting tasks have been relatively well discussed previously [7,8]. However, studies of psychophysical factors in this realm were only found with few reports. There is no doubt that a comprehensive understanding of the effect of psychophysical factors in manual lifting could help practitioners in occupational safety and health upgrade intervention programs to minimize the occurrence of LBP.

Psychophysical factors, which reflect perceived physical demand in lifting tasks, were found to interact with biomechanical factors to change body movement patterns [9,10]. Kujala et al. (1996) found that higher perceived physical demand responded to higher load stress because of the increased tension of muscular contraction. When people were doing difficult lifting tasks (e.g., higher lifting frequency, asymmetric lifting), perceived physical demand increased [10,11]. Perceived physical demand was even affected by load knowledge such as weight information of the handled target. Studies on load knowledge generally concluded that availability of load knowledge decreased low back physical stress in the lifting task [12,13,14]. These studies particularly pointed out that the kinetic performance of the low back was improved when people was given load weight knowledge of the handled target compared with having no load weight knowledge.

Although the interaction effect between biomechanical and psychophysical factors has been studied, questions regarding the influence of individual psychophysical capacity, load knowledge, and other occupational factors in task settings on low back biomechanics were still left unanswered. First, the load weight setting in the lifting task was arbitrary in many previous studies and could not reflect individual lifting capacity. Second, there were more than just one form of load knowledge for the handled target. Workers would often handle targets with fragile content. Third, previous results were all found at the lifting phase of the lifting task. A manual lifting period in daily routine work comprises successive lifting and deposit phases. No one has reported if the previous finding is consistent throughout the lifting period. Last, asymmetric lifting is more dangerous in terms of developing LBP than symmetric lifting [15]. Low back biomechanics with known load knowledge has not been observed in existing studies for asymmetric lifting tasks.

The purpose of this study was to provide additional information about the influence of psychophysical lifting load weight and load knowledge on lifting biomechanics in the asymmetric lifting task. According to the individual psychophysical lifting capacity measurement, low, medium, and high lifting load weights were selected. Load knowledge of ‘no knowledge (NK)’, ‘weight known (WK)’, ‘fragile material known (FK), and ‘weight and fragile material known (WFK)’ were given for examination. The hypothesis of this study was that load knowledge could decrease the low back physical stress during the entire lifting period, and there would be interaction effects between load knowledge and lifting load weight (i.e., changes of lifting biomechanics between load knowledge was different at different levels of lifting load weight).

## 2. Materials and Methods

### 2.1. Participants

To calculate the sample size, the G*Power software version 3.0.10 was used [16]. To detect a medium effect size of 0.3 with a statistical power of 80% and a level of significance of 5%, a minimal sample size of 18 was required based on the data from Farrag et al. [13]. Twenty-four male participants (age: 36.2 ± 8.5 years; height: 170.6 ± 4.6 cm; weight: 74.9 ± 9.1; knuckle height: 75.7 ± 2.7 cm) were recruited from the local community by using online advertisement and posters. The inclusion criterion of the participants was male adults who had manual material lifting tasks in their daily job in the last six months. The exclusion criteria were having any low back injury history and not being capable of finishing the psychophysical test. Informed consent was obtained from all the participants. The Institutional Review Board of Shenzhen University has approved this study.

### 2.2. Instrumentation

An eight-camera optoelectronic system sampling at 100 Hz was applied to measure lifting kinematics and the movement trajectories of lifting load. Thirty-nine reflective markers were placed on anatomic body landmarks according to the Vicon standard procedure for establishing the plug-in gait (PIG) full-body model (Plug-In Gait Marker Set, Vicon Peak, Oxford, UK). An extra four reflective markers were taped on the top corners of the box. To acquire the lifting kinetics, two force plates (FP4060, Bertec, Columbus, OH, USA) with a sampling rate of 1000 Hz were paralleled positioned on the ground to collect ground reaction force. Synchronization of the motion capture system and force plates was done by setting the signals to start simultaneously.

### 2.3. Experiment Procedure

The experiment was designed to have two sessions with an interval of one week. In the first session, the participant’s maximum acceptable lifting capacity in an asymmetric lifting task was measured by using a psychophysical test. The psychophysical test was completed in a simulated work environment. A box (40 cm × 26 cm × 26 cm) was lifted from the ground to a table of knuckle height located asymmetrically 90° at the left of the sagittal plane of the participant (Figure 1). The lifting pace was four lifts per minute. Participants were first asked to adjust the standing position relative to the table and box to perform a comfortable lifting posture. An empty box was given to the participant during the adjustment to practice the simulated lifting. The feet position, table position, and horizontal location of box were recorded for the participant to use in the psychophysical test and the lifting test one week later.

When the psychophysical test started, participants lifted a relatively small initial load weight (i.e., 2 kg). Load weight with a differential of 1 kg in ascending order for each lifting trial was provided to participants. Participants did each lifting trial and were asked to feedback if the load weight was acceptable for an eight-hour work shift. The ascending trials were stopped once the participants self-reported that load was unacceptable for an eight-hour work shift. Subsequently, after a short break, participants were instructed to perform descending lifting trials. A relatively large initial load weight (i.e., 20 kg) was used, and load weight with a differential of 1 kg in descending order for each lifting trial was provided to participants until participants reported the load weight to be acceptable for an eight-hour work shift. The average of the two transition load weights (9.0 ± 3.2 kg) was used as the maximum acceptable lifting capacity.

In the second session, one week later, biomechanical data were collected when participants performed asymmetric lifting tasks under different lifting load weight and load knowledge conditions. The asymmetric lifting task was similar to that specified in the psychophysical test. In particular, participants were instructed to transfer a box with handles on sides from the ground to an asymmetric table of knuckle height located at 90° to the left of sagittal plane of the participant at their self-selected pace. Testing conditions were defined by different lifting load weights and load knowledge levels. The lifting load weight was set at three levels, corresponding to 40% (low), 80% (medium), and 120% (high) of the maximum acceptable lifting capacity, respectively. The factor of load knowledge had four levels corresponding to ‘no knowledge (NK)’, ‘weight known (WK)’, ‘fragile material known (FK)’, and ‘weight and fragile material known (WFK)’ conditions. Load weight and fragility information was provided to participants using a label attached to the top of the box (Figure 2). A factorial design was adopted. Therefore, there were 12 testing conditions (3 load weight conditions × 4 load knowledge conditions) in the experiment, and the sequence of these testing conditions was randomly arranged by the lifting load weight. A Latin square design was used for the load knowledge under each lifting load weight across participants. Participants were asked to do three lifting trials with at least one minute between two consecutive trials in each testing condition.

### 2.4. Data Processing and Dependent Variables

Due to marker missing, three participants were not included in further data analysis. Raw motion capture data were digitally filtered using the Woltring filtering MSE routine with a cut-off frequency of 10 Hz. The fourth-order Butterworth filter processed raw force plate signals with zero lag.

Dependent variables reflecting low back biomechanics were defined by trunk kinematics and low back load. The trunk angular displacement has been reported to significantly influence the low back loading [17]. Trunk angular velocity was used to distinguish high-risk and low-risk lifting tasks [18]. In addition, trunk angular acceleration could further explain the spinal loading affected by trunk kinematic performance [19]. The trunk angle, angular velocity, and acceleration in the sagittal, transversal, and horizontal planes were thus chosen. Their peak values in the lifting and deposit phases were selected as dependent variables. The lifting phase and deposit phase were separated when the box was close to the body. The reference trunk posture (i.e., 0 degrees) was defined at the standard anatomical position, and positive joint angles suggested sagittal flexion, transverse twisting, and lateral flexion. The moment at L5–S1 joint has been used as the indicator of risk factor of LBP [20]. Thus, the peak L5–S1 moment observed in the whole lifting period and the average L5–S1 moment from the lifting and deposit phases were taken as the dependent variables accounting for the low back load. The kinetic performance of lumbar disks L5–S1 was calculated based on the modified plug-in gait model called S-model [21]. The force and moment of the joint L5–S1 were developed by inverse dynamics algorithms using ground reaction forces collected from the force plates and body kinematics collected from the motion capture system.

### 2.5. Statistical Analysis

This study used a two-factor within-subject repeated-measures design. The main effects of load knowledge and lifting load weight and the effects of their interactions were examined by a repeated-measures ANOVA. Data normal distribution was visually checked using the residual plots technique under each group of the 12 combinations of the examined factors (i.e., load weight and load knowledge). If any data point was larger than three times of standard deviation from the group mean, the corresponding participant was removed from the data analysis. Thus, another two subjects were excluded by this criterion. Nineteen participants were finally analyzed in this study. The Mauchly sphericity test was used to test the assumption of sphericity. Greenhouse–Geisser epsilon estimate was used to correct the degree of freedom for critical F-value when the sphericity assumption was violated. When a significant main effect (*p* < 0.05 was found for either load knowledge or lifting load weight, post-hoc analysis was carried out for pairwise comparisons using the Tukey procedure. A significant interaction effect (*p* < 0.05) was found between the two examined factors, pairwise comparisons were made between load knowledge conditions at the levels of low, medium, and high lifting load weight, separately, using paired *t*-tests. All statistical analyses were done in R statistical software version 4.0.5 (R. Core Team, Vienna, Austria).

## 3. Results

In this study, no interaction effect between load knowledge and lifting load weight was found. Significant main effects of load knowledge were only located at the deposit phase. Although there were significant main effects for lifting load weight at both lifting phase and deposit phase, the effects can be only seen on variables of trunk average sagittal bending moment.

In terms of the main effects of load knowledge at the deposit phase, trunk sagittal flexion velocity decreased significantly from WK condition to WFK condition. Additionally, lateral flexion velocity and acceleration were significantly lowered when load fragility information was presented (i.e., WFK and FK conditions) compared with NK. Lifting kinetic performance showed a significant difference of low back average lateral bending moment between WK and FK conditions (Table 1).

In terms of the main effect of lifting load weight, a significant difference in trunk transverse twisting velocity was observed between medium and high load weights at the lifting phase. In addition, there was a significant difference in trunk sagittal flexion velocity between low and high load weights. Unlike load knowledge, lifting load weight significantly affected peak low back moments in all three-dimensional planes: the higher lifting load weight, the higher peak low back moment. The average sagittal bending moment at the lifting phase and average moment in all three-dimensional planes at the deposit phase significantly rose with increased lifting load weight (Table 2).

## 4. Discussion

The main purpose of this study was to examine the load knowledge effects on lifting biomechanics during asymmetric lifting. Participants were instructed to perform asymmetric lifting under four load knowledge conditions when handling different levels of load weight defined by their maximum acceptable lifting capacity. Lifting biomechanics was assessed using trunk kinematics and low back moments.

It was found that trunk kinematics measured in the WK condition was similar to that in the NK condition. This result was in disagreement with previous reports that trunk kinematic load was significantly higher under NK than under WK [22]. For example, a previous study by Kotowski et al. reported larger trunk velocities in the sagittal and lateral planes when lifting unknown load weight by male participants. The discrepancy could be explained by the different experimental protocols adopted between Kotowski et al. and ours. Specifically, Kotowski et al. set lifting frequency at 8 lifts/min with different load weight levels presented in a random sequence. Even though we also applied the random sequence for lifting load weight levels, participants did not give instruction on lifting frequency. This allowed participants to have enough time to adjust their perturbed trunk movement when the actual weight was different from the estimated weight at the NK. The movement strategy adjustment could happen at the grasping phase. There was no difference in lifting strategies at both lifting and deposit phases regardless of the knowledge of load weight.

Previous studies reported that lifting light weight with known mass can decrease the low back load when it was compared with the condition without knowing load weight [12,13]. The present study disagreed with the previous finding as well since no difference in low back kinetic load between NK and KW was found. It has been shown that the postural adjustment induced by the unexpected load could be a factor affecting the low back load under the no load knowledge condition [23]. However, in this study, participants were informed at the beginning of the experiment that they would be exposed to different levels of load weight in random order. Thus, they might not expect to lift the same weight at the time of the next lift in the NK condition, and thus possibly adjusted their lifting strategy at the grasping phase while remaining the similar strategy at the lifting phase in the following NK trial. This adjustment could be similar to that they made in the KW condition, as evidenced by the observed similar trunk joint angles between the NK and KW conditions at both the lifting phase and deposit phase. Low backload is directly influenced by the trunk joint angles [24].

Providing the fragility information of a load is commonly used as a precaution for people to handle objects containing fragile materials such as porcelain. However, to our knowledge, this is the first study to compare the asymmetric lifting biomechanics between the conditions of knowing material fragility and no-load knowledge. We found that load knowledge of material fragility influenced the asymmetric lifting biomechanics at the deposit phase. Specifically, participants lowered their trunk lateral flexion velocity and acceleration when the handled object was presented with load fragility information in asymmetric lifting. This lifting strategy can be explained as an attempt to avoid breaking the fragile materials. When a fragile object is required to be lifted, slowed movement can avoid the jerk reaction [25], which may prevent the damage of the object. We also observed significantly lowered trunk sagittal flexion velocity and trunk average lateral bending moment in the KF condition compared with the KW condition. This can further support the idea that fragility information has led to a different cognitive process and changed the lifting strategy. From the perspective of LBP prevention, our findings imply that load knowledge of material fragility could lower the risk of developing the LBP in the asymmetric lifting task.

Consistent with the previous study, lifting load weight has been identified as a determinant factor of low back (L5–S1) load [24], as the low back peak moment on all three-dimensional planes were significantly increased with larger lifting load weight. Song and Qu (2014) have reported significantly higher average low back moment in both the lifting phase and deposit phase of asymmetric lifting with increased lifting load weight [19]. However, in this study, the changes on the three-dimensional planes were only found at the deposit phase. This suggests that lifting load weight in the asymmetric task affected the low back load the most at the deposit phase.

Previous studies have pointed out that there is an interaction effect between load knowledge and lifting load weight on lifting biomechanics in the symmetric lifting task [12,13,14,22,26]. Lifting biomechanics such as low back posture, low back acceleration and low back load were lowered under known weight conditions with light lifting load weight compared with no load knowledge condition [12,13,14,22,26]. However, no interaction effect between load knowledge and lifting load weight on asymmetric lifting biomechanics was found in this study. This is possibly due to the asymmetric lifting task is more physically demanding than the symmetric lifting task, so participants encountered difficult postural control situations to change their body movement patterns under each level of lifting load weight when load knowledge condition changes. Even though the lifting load weight in this study was designed by the maximum acceptable lifting capacity, the load set at 40% of the maximum acceptable lifting capacity could also be perceived as not easy for asymmetric lifting.

The results of decreased trunk lateral flexion velocity and acceleration implied that load knowledge such as fragility instead of the perceived lifting load weight can lead to a more cautious lifting strategy to lower the risk of LBP during the asymmetric lifting task. However, several factors such as the lifting frequency, destination height, and lifting load size may limit the generalization of this study. In addition, only participants with manual material lifting experience were recruited. The impact of the fragility load knowledge on common people is not guaranteed. In addition, there is a need to do an injury prevention study on LBP in the real working environment to test the implication of this study.

## 5. Conclusions

In summary, after the kinematic and kinetic data from nineteen participants doing asymmetric tasks were analyzed, we found a decreased trunk lateral flexion velocity and acceleration when the lifting load weight was presented with fragility knowledge. The finding remained regardless of the perceived lifting load weight obtained from the psychophysical test. Even though trunk postural angle, a kinematic factor closely related to low back load, was not affected by material fragility, the improved trunk kinematic performance suggests providing material fragility information to workers in asymmetric lifting tasks be effective in reducing their risk of LBP. This study has enriched our knowledge regarding the influence of load knowledge on asymmetric lifting biomechanics and the risk of LBP. Load knowledge, such as fragility, can change the strategy of human movement regardless of the lifting load weight in the asymmetric lifting task. It would help a practitioner to develop the LBP prevention protocol for workers having manual material lifting tasks in their daily tasks.

## Figures and Tables

**Figure 1 ijerph-19-03207-f001:**
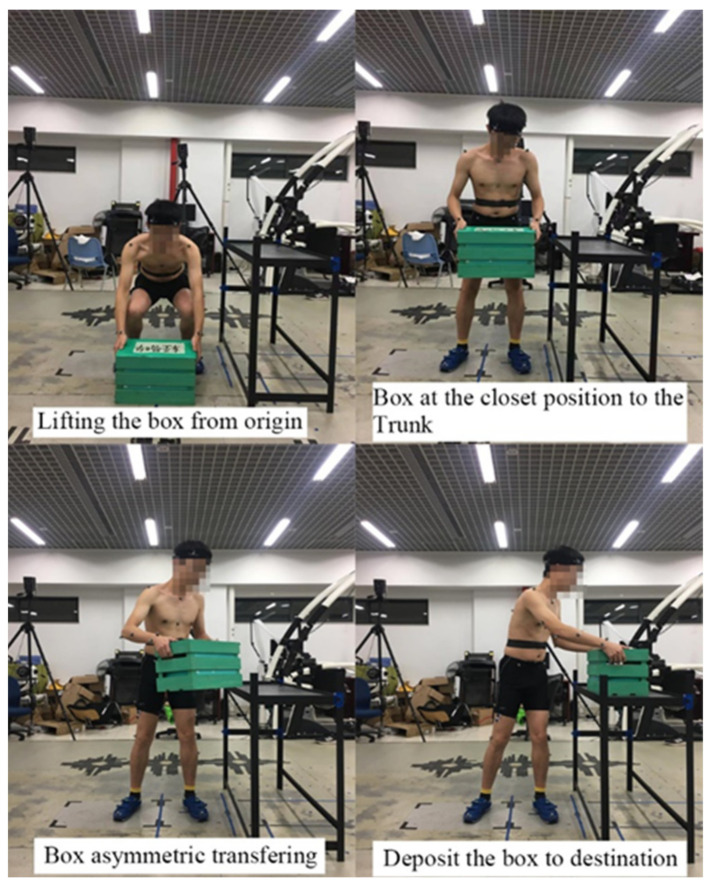
A participant performing the asymmetric lifting task.

**Figure 2 ijerph-19-03207-f002:**
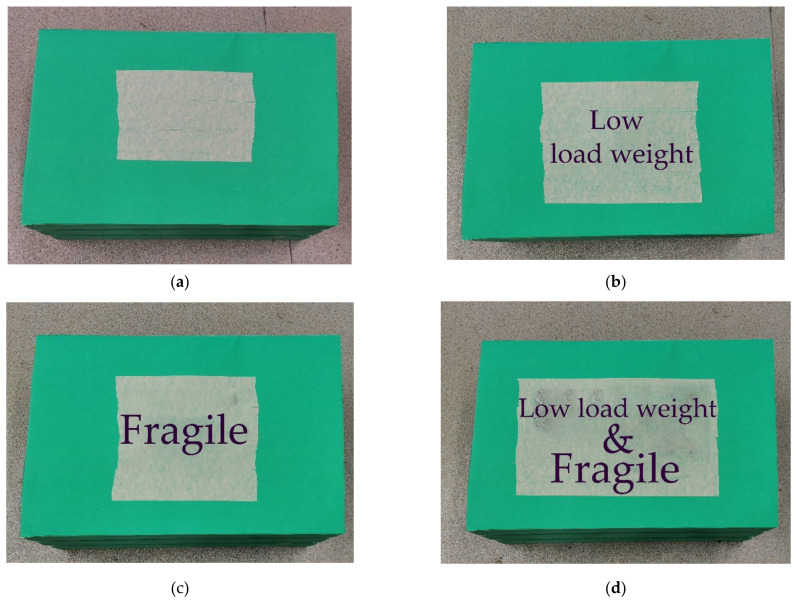
The lifting boxes used in each load knowledge condition. (**a**) NK: no knowledge; (**b**) WK: weight known; (**c**) FK: fragile material known; (**d**) WFK: weight and fragile material known.

**Table 1 ijerph-19-03207-t001:** Main effects of load knowledge: Mean ± SD.

	NK	WK	WFK	FK	*p*
**Peak Trunk Kinematics (Lifting phase)**					
Sagittal flexion angle (°)	47.7 ± 2.5	47.9 ± 2.5	48.2 ± 2.6	48.7 ± 2.4	0.386
Sagittal flexion velocity (°/s)	40.9 ± 2.2	43.3 ± 3.2	41.9 ± 3.0	43.1 ± 3.5	0.573
Sagittal flexion acceleration (°/s^2^)	239.8 ± 20.0	241.5 ± 22.3	221.7 ± 17.3	239.1 ± 21.3	0.224
Transverse twisting angle (°)	3.0 ± 0.3	3.3 ± 0.3	3.2 ± 0.3	3.0 ± 0.3	0.130
Transverse twisting velocity (°/s)	9.5 ± 1.1	9.7 ± 0.8	9.3 ± 0.9	9.7 ± 0.8	0.898
Transverse twisting acceleration (°/s^2^)	71.1 ± 9.5	78.3 ± 10.0	70.3 ± 9.8	69.6 ± 7.4	0.394
Lateral flexion angle (°)	3.4 ± 0.3	3.6 ± 0.4	3.4 ± 0.3	3.3 ± 0.3	0.554
Lateral flexion velocity (°/s)	10.9 ± 1.0	11.1 ± 0.9	10.3 ± 0.9	10.3 ± 0.9	0.536
Lateral flexion acceleration (°/s^2^)	118.8 ± 17.6	120.0 ± 14.9	105.0 ± 14.5	109.2 ± 14.8	0.155
**Peak Trunk Kinematics (Deposit Phase)**					
Sagittal flexion angle (°)	37.1 ± 2.4	37.0 ± 2.3	36.8 ± 2.5	36.6 ± 2.3	0.822
Sagittal flexion velocity (°/s)	55.8 ± 4.3	57.1 ± 4.7	53.9 ± 4.1	55.1 ± 4.4	0.077
Sagittal flexion acceleration (°/s^2^)	210.5 ± 22.4	208.3 ± 22.4 ^‡^	186.0 ± 17.8	199.1 ± 20.3	**0.029**
Transverse twisting angle (°)	14.3 ± 0.8	14.1 ± 0.8	14.2 ± 0.9	14.6 ± 0.9	0.479
Transverse twisting velocity (°/s)	18.8 ± 0.9	18.7 ± 1.0	18.2 ± 1.1	18.6 ± 1.1	0.806
Transverse twisting acceleration (°/s^2^)	102.9 ± 10.1	96.4 ± 8.6	90.6 ± 8.3	92.6 ± 9.2	0.103
Lateral flexion angle (°)	19.7 ± 0.8	20.0 ± 0.9	19.5 ± 0.9	19.6 ± 0.9	0.324
Lateral flexion velocity (°/s)	37.3 ± 2.0 *^†^	37.0 ± 2.2	34.2 ± 1.9	35.2 ± 2.2	**0.005**
Lateral flexion acceleration (°/s^2^)	189.2 ± 19.7 *^†^	177.7 ± 14.3	162.1 ± 15.6	164.6 ± 14.9	**0.010**
**Low Back Moment (L5–S1)**					
Peak sagittal bending moment (N·m)	225.4 ± 8.3	225.1 ± 8.4	222.8 ± 8.4	226.0 ± 8.9	0.123
Peak lateral bending moment (N·m)	47.4 ± 4.5	47.1 ± 4.3	46.4 ± 4.3	45.1 ± 4.4	0.066
Peak twisting moment (N·m)	25.4 ± 2.9	24.8 ± 2.3	25.2 ± 2.5	24.5 ± 2.4	0.641
Average sagittal bending moment—Lifting phase (N·m)	193.7 ± 8.1	192.5 ± 7.6	191.4 ± 8.0	193.1 ± 8.3	0.596
Average lateral bending moment—Lifting phase (N·m)	6.2 ± 0.7	6.5 ± 0.6	6.3 ± 0.7	6.1 ± 0.6	0.743
Average twisting moment—Lifting phase (N·m)	9.1 ± 1.9	7.6 ± 0.9	8.2 ± 1.0	8.0 ± 0.9	0.517
Average sagittal bending moment—Deposit phase (N·m)	74.0 ± 4.6	72.6 ± 4.6	72.9 ± 4.8	73.4 ± 5.0	0.567
Average lateral bending moment—Deposit phase (N·m)	23.4 ± 2.4	23.4 ± 2.3 ^§^	23.0 ± 2.4	22.4 ± 2.3	**0.039**
Average twisting moment—Deposit phase (N·m)	12.4 ± 1.4	12.5 ± 1.5	12.6 ± 1.5	12.3 ± 1.4	0.782

Note: * Indicates significant difference between the NK and FK conditions. ^†^ indicates a significant difference between the NK and WFK conditions. ^‡^ indicates significant difference between the WK and WFK conditions. ^§^ indicates significant difference between the WK and FK conditions.

**Table 2 ijerph-19-03207-t002:** Main effects of load weight: Mean ± SD.

	Low	Medium	High	*p*
**Peak Trunk Kinematics (Lifting Phase)**				
Sagittal flexion angle (°)	48.2 ± 2.6	47.3 ± 2.7	48.9 ± 2.2	0.119
Sagittal flexion velocity (°/s)	46.5 ± 3.4	40.4 ± 3.0	40.0 ± 3.3	0.072
Sagittal flexion acceleration (°/s^2^)	243.1 ± 23.5	226.6 ± 18.2	236.8 ± 20.8	0.493
Transverse twisting angle (°)	3.3 ± 0.3	3.1 ± 0.3	3.0 ± 0.3	0.433
Transverse twisting velocity (°/s)	10.0 ± 0.9	10.0 ± 0.9 ^‡^	8.7 ± 0.8	**0.024**
Transverse twisting acceleration (°/s^2^)	72.2 ± 9.4	74.9 ± 8.4	69.8 ± 8.9	0.476
Lateral flexion angle (°)	3.3 ± 0.3	3.4 ± 0.4	3.5 ± 0.3	0.670
Lateral flexion velocity (°/s)	10.8 ± 1.0	10.5 ± 0.9	10.6 ± 0.8	0.816
Lateral flexion acceleration (°/s^2^)	113.1 ± 15.8	111.6 ± 15.7	115.0 ± 14.5	0.827
**Peak Trunk Kinematics (Deposit Phase)**				
Sagittal flexion angle (°)	36.5 ± 2.4	36.3 ± 2.7	37.8 ± 2.3	0.270
Sagittal flexion velocity (°/s)	58.2 ± 5.0 ^†^	53.9 ± 4.4	54.4 ± 3.8	**0.025**
Sagittal flexion acceleration (°/s^2^)	200.0 ± 19.5	197.5 ± 20.8	205.4 ± 22.3	0.595
Transverse twisting angle (°)	14.2 ± 0.9	14.3 ± 0.8	14.3 ± 0.9	0.944
Transverse twisting velocity (°/s)	18.6 ± 1.0	19.1 ± 1.1	18.0 ± 0.9	0.077
Transverse twisting acceleration (°/s^2^)	93.2 ± 9.0	94.1 ± 8.3	99.7 ± 11.4	0.559
Lateral flexion angle (°)	19.8 ± 1.0	19.9 ± 0.8	19.4 ± 0.9	0.412
Lateral flexion velocity (°/s)	35.2 ± 1.9	36.1 ± 2.3	36.5 ± 1.9	0.263
Lateral flexion acceleration (°/s^2^)	169.6 ± 13.5	171.1 ± 17.8	179.5 ± 15.9	0.101
**Low Back Moment (L5–S1)**				
Peak sagittal bending moment (N·m)	205.9 ± 7.4 *^†^	224.4 ± 8.6 ^‡^	244.1 ± 9.8	**<0.001**
Peak lateral bending moment (N·m)	35.2 ± 3.1 *^†^	47.6 ±4.5 ^‡^	56.7 ± 5.7	**<0.001**
Peak twisting moment (N·m)	20.9 ± 2.1 *^†^	24.7 ± 2.4 ^‡^	29.3 ± 3.1	**<0.001**
Average sagittal bending moment—Lifting phase (N·m)	175.4 ± 7.3 *^†^	192.7 ± 8.2 ^‡^	210.0 ± 8.8	**<0.001**
Average lateral bending moment—Lifting phase (N·m)	6.6 ± 0.8	6.0 ± 0.6	6.2 ± 0.6	0.387
Average twisting moment—Lifting phase (N·m)	8.5 ± 1.4	7.6 ± 0.8	8.5 ± 1.0	0.540
Average sagittal bending moment—Deposit phase (N·m)	57.7 ± 3.8 *^†^	73.2 ± 4.6 ^‡^	88.8 ± 6.0	**<0.001**
Average lateral bending moment—Deposit phase (N·m)	17.7 ± 1.7 *^†^	23.5 ± 2.4 ^‡^	28.0 ± 3.0	**<0.001**
Average twisting moment—Deposit phase (N·m)	10.1 ± 1.2 ^*†^	12.9 ± 1.5 ^‡^	14.4 ± 1.7	**<0.001**

Note: * Indicates significant difference between low and high load weight conditions. ^†^ indicates significant difference between low and medium load weight conditions. ^‡^ indicates significant difference between medium and high load weight conditions.

## Data Availability

The data are available from the corresponding author upon reasonable request.

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
