# Peer review of "Influence of Load Knowledge on Biomechanics of Asymmetric Lifting"

_ijerph, 2022, doi:10.3390/ijerph19063207_

Round 1
Reviewer 1 Report
The study is very interesting in that it shows the effect of knowledge of load on lifting kinematics as well as kinetics. Very few studies have spent effort in this research area. Comments are below.
(1) Please explain more details of the calculation on lumbar disks L5-S1.
(2) Please explain a bit more details for how to prevent learning effects in the experiment.
(3) Please comment on why some variables have no significant differences across conditions (low - high). e.g. Sagittal flexion angle - Peak Trunk Kinematics (Lifting Phase)
Author Response
The authors of this paper really appreciate the reviewer’s comments. Pleae find our point-by-point response to the reviewer's comments below.
Comment #1
(1) Please explain more details of the calculation on lumbar disks L5-S1.
Response:
Thanks for the comment. More details have been provided in the revised manuscript as follows.
“The kinetic performance of lumbar disks L5-S1 was calculated based on the modified plug-in gait model called S-model [21]. The force and moment of the joint L5-S1 were developed by inverse dynamics algorithms using ground reaction forces collected from the force plates and body kinematics collected from the motion capture system.” (Line 163-167)
Comment #2
(2) Please explain a bit more details for how to prevent learning effects in the experiment.
Response:
We appreciate the reviewer’s comment. The lifting sequence was randomized to prevent learning effects. Relevant revision has been made in the paper and provided below.
“Therefore, there were in total 12 testing conditions (3 load weight conditions × 4 load knowledge conditions) in the experiment, and the sequence of these testing conditions was randomly arranged by the lifting load weight and a Latin square design was used for the load knowledge under each lifting load weight across participants.” (Line 136-140)
Comment #3
(3) Please comment on why some variables have no significant differences across conditions (low - high). e.g. Sagittal flexion angle - Peak Trunk Kinematics (Lifting Phase)
Response:
We appreciate the reviewer’s comment. The finding of no significant differences of peak trunk kinematics across conditions (low-high) agreed with a previous study (Song & Qu, 2014), in which the lifting load weights were set at 5%, 15%, and 25% of participants’ maximum lifting capacity corresponding to 1.2 kg, 3.6 kg, and 6.0 kg on average, respectively. No significant difference was observed between 15% (3.6kg) and 25% (6.0kg) weight conditions. In the current study, the lifting load weight was set with an average of 3.5 kg, 7.0 kg, and 10.5 kg for low, medium, and high lifting weights, respectively, and no significant differences of peak trunk kinematics was found between load weight conditions.
Reviewer 2 Report
Influence of Load Knowledge on Biomechanics of Asymmetric 2 Lifting
- Materials and Methods
|
How have you calculated the study sample? You must specify
You define the study population as: · Male participants: (age: 36.2 ± 8.5 years; height: 170.6 ± 4.6 cm; weight: 77 74.9 ± 9.1; knuckle height: 75.7 ± 2.7 cm) · Recruited from the local community
· What are the inclusion criteria?
· What are the characteristics of the population in which the study was conducted?
Is this the only exclusion criteria? Why?
· Without any low back injury history.
It's confusing, what is your "N" finally in this study?
· […] five participants were not included in further data analysis […]
· […] If any data point was larger than 3 times of standard deviation from the group mean, the corresponding participant was removed from the data analysis. Thus, another two subjects were excluded by this criterion. […] |
76-82
138
161-163 |
- Discussion
|
What limitations have you found in your study? |
203 |

Author Response
The authors of this paper really appreciate the reviewer’s comments. Pleae find our point-by-point response to the reviewer's comments below.
Comment #1
(1) How have you calculated the study sample? You must specify
Response:
We appreciate the reviewer’s comment. Additional explanation has been added in the paper with the following information.
“To calculate the sample size, the G*Power software version 3.0.10 was used [16]. To detect a medium effect size of 0.3 with a statistical power of 80% and a level of significance of 5%, a minimal sample size of 18 was required based on the data from Farrag et al. [13].” (Line 77 – 79)
Comment #2
(2) What are the inclusion criteria?
Response:
Thanks for the comment and we have put inclusion criterion in the relevant places in the paper.
“The inclusion criterion of the participants was male adults who had manual material lifting tasks in their daily job in the last six months.” (Line 83 – 85)
Comment #3
(3) What are the characteristics of the population in which the study was conducted?
Response:
We appreciate the reviewer’s comment. A convenience sample was recruited by the online advertisement and posters given to the local community. The characteristics of the population was determined by the inclusion criteria and exclusion criteria. Details of the inclusion criteria and exclusion criteria were provided below.
“The inclusion criterion of the participants was male adults who had manual material lifting tasks in their daily job in the last six months. The exclusion criteria were having any low back injury history and not capable of finishing the psychophysical test.” (Line 83 – 86)
Comment #4
(4) Is this the only exclusion criteria? Why? Without any low back injury history.
Response:
Thanks for the comment. The exclusion criteria have been specifically described as follows in the revision.
“The exclusion criteria were having any low back injury history and not capable of finishing the psychophysical test.” (Line 85 – 86)
Comment #5
(5) what is your "N" finally in this study?
Response:
Thanks for the comment. The information about the exact number of participants used for analysis has been provided as follows.
“Due to marker missing, three participants were not included in further data analysis.” (Line 146 - 147)
“Nineteen participants were finally analyzed in this study.” (Line 175 – 176)
Comment #6
(6) What limitations have you found in your study?
Response:
We appreciate the reviewer’s comment. There were some limitations we thought might limit the generalization of this study and they are provided in the last paragraph in the discussion part.
“… several factors such as the lifting frequency, destination height, and lifting load size may limit the generalization of this study. In addition, only participants with manual material lifting experience were recruited. The impact of the fragility load knowledge on common people is not guaranteed. Also, there is a need of doing injury prevention study on LBP in the real working environment to test the implication of this study.” (Line 291 – 296)
Reviewer 3 Report
Reviewer’s comments
Manuscript Number: IJERPH - 1611163
The purpose of this study was to provide additional information about the influence of psychophysical lifting load weight and load knowledge on lifting biomechanics in the asymmetric lifting task.
The manuscript is well written and organized, but the following comments and suggestions will help the authors to improve the manuscript:
--The authors provided a structured abstract including (1) background; (2) methods. Please look at the journal’s website for authors’ instruction to guide the structure of writing an abstract
--There are numbered mentioned to the keywords. Kindly remove them
--The reviewer wasn’t sure of where the force plates were used during the experimental procedure
-- What measurement parameters were derived from the force plates during data processing
-- Please state what statistical tool (e.g., SPSS) was used for statistical analyses
--Before the “conclusions” section, the authors must add another section to discuss the “study implication, limitations and future research directions”
--The “conclusions” section was poorly written. It must summarize the research objective, research methods, key results, findings and implication, contributions.

Author Response
The authors really appreciate the reviewer's comments. Please find our point-by-point response to the reviewer's comments below.
Comment #1
The authors provided a structured abstract including (1) background; (2) methods. Please look at the journal’s website for authors’ instruction to guide the structure of writing an abstract.
Response:
We appreciate the reviewer’s comment. The abstract has been rewritten as follows to meet the journal’s requirement.
“(1) Background: Load knowledge has been identified as a factor affecting the risk of low back pain (LBP) during symmetric lifting. However, the effects of load knowledge in asymmetric lifting tasks have not been reported yet. The purpose of this study was to investigate the load knowledge influence on lifting biomechanics in asymmetric lifting tasks; (2) Methods: Twenty-four male adults were recruited to complete a psychophysical lifting capacity test and a simulated asymmetric lifting task. The lifting task was set with load knowledge of ‘no knowledge’ (NK), ‘weight known’ (WK), ‘fragile material known’ (FK) and ‘weight and fragile material known’ (WFK) for different lifting load weights. Trunk kinematics and kinetics were collected and analyzed; (3) Results: When fragility information was presented, trunk sagittal flexion acceleration, lateral flexion velocity and acceleration, and average lateral bending moment were significant lowered at the deposit phase. Lifting high load weight was found to significantly increase low back sagittal bending moment at the lifting phase and low back moments of all three-dimensions at the deposit phase; (4) Conclusions: The decrease of trunk kinematic load suggests that providing material fragility information to workers in asymmetric lifting task would be effective in reducing their risk of LBP.” (Line 9 – 22)
Comment #2
There are numbered mentioned to the keywords. Kindly remove them.
Response:
Thanks for the reviewer’s comment. The numbers have been removed from the Keywords.
“Keywords: Biomechanics; asymmetric lifting; load knowledge; low back pain.” (Line 23)
Comment #3
The reviewer wasn’t sure of where the force plates were used during the experimental procedure.
Response:
We appreciate the reviewer’s comment. There were two tests in this study. The first test was the psychophysical test. The second test was the test with the load knowledge. The force plates were used only during the second test to collect the ground reaction forces and moments which were further used to calculate low back force and moments via inverse dynamics algorithms.
Comment #4
What measurement parameters were derived from the force plates during data processing?
Response:
Thanks for the reviewer’s comment. In this study, the measurement parameters derived from the force plates were three-dimensional ground reaction forces and moments.
Comment #5
Please state what statistical tool (e.g., SPSS) was used for statistical analyses.
Response:
We appreciate the reviewer’s comment. Relevant information has been added and put in the paper with green text.
“All statistical analyses were done in R-statical software version 4.0.5 (R. Core Team, Austria).” (Line 184 - 185)
Comment #6
Before the “conclusions” section, the authors must add another section to discuss the “study implication, limitations and future research directions”
Response:
Thanks for the reviewer’s comment. We have provided an extra paragraph at the end the discussion to discuss “study implication, limitations and future research directions” according to the reviewer’s suggestion.
“The results of decreased trunk lateral flexion velocity and acceleration implied that this study implied that load knowledge such as fragility instead of the perceived lifting load weight can lead to a more cautious lifting strategy to lower the risk of LBP during the asymmetric lifting task. However, several factors such as the lifting frequency, destination height, and lifting load size may limit the generalization of this study. In addition, only participants with manual material lifting experience were recruited. The impact of the fragility load knowledge on common people is not guaranteed. Also, there is a need of doing injury prevention study on LBP in the real working environment to test the implication of this study.” (Line 288 – 296)
Comment #7
The “conclusions” section was poorly written. It must summarize the research objective, research methods, key results, findings and implication, contributions.
Response:
We appreciate the reviewer’s comment. The ‘conclusions’ section has been rewritten as follows.
“In summary, after the kinematic and kinetic data from nineteen participants doing asymmetric tasks were analyzed, we found a decreased trunk lateral flexion velocity and acceleration when the lifting load weight was presented with fragility knowledge. The finding remained regardless of the perceived lifting load weight obtained from the psychophysical test. Even though trunk postural angle, which is a kinematic factor closely related to low back load, was not affected by the knowledge of material fragility, the improved trunk kinematic performance suggests that providing material fragility information to workers in asymmetric lifting tasks would be effective in reducing their risk of LBP. This study has enriched our knowledge regarding the influence of load knowledge on asymmetric lifting biomechanics and further on the risk of LBP. Load knowledge such as fragility can change the strategy of human movement regardless of the lifting load weight in the asymmetric lifting task. It would help practitioner to develop the LBP prevention protocol for workers having manual material lifting tasks in their daily job.” (Line 298 – 310)
Round 2
Reviewer 3 Report
Reviewer’s comments
Manuscript Number: IJERPH – 1611163V2
The authors have thoroughly and satisfactorily addressed the review comments. I recommend its publication.